# Charge carrier mapping for Z-scheme photocatalytic water-splitting sheet via categorization of microscopic time-resolved image sequences

Makoto Ebihara[1], Takeshi Ikeda[2,3], Sayuri Okunaka[2,3], Hiromasa Tokudome [2,3], Kazunari Domen [4,5] & Kenji Katayama [1✉]

Photocatalytic water splitting system using particulate semiconductor materials is a promising strategy for converting solar energy into hydrogen and oxygen. In particular, visible-light-driven 'Z-scheme' printable photocatalyst sheets are cost-effective and scalable. However, little is known about the fundamental photophysical processes, which are key to explaining and promoting the photoactivity. Here, we applied the pattern-illumination time-resolved phase microscopy for a photocatalyst sheet composed of Mo-doped $BiVO_4$ and Rh-doped $SrTiO_3$ with indium tin oxide as the electron mediator to investigate photo-generated charge carrier dynamics. Using this method, we successfully observed the position- and structure-dependent charge carrier behavior and visualized the active/inactive sites in the sheets under the light irradiation via the time sequence images and the clustering analysis. This combination methodology could provide the material/synthesis optimization methods for the maximum performance of the photocatalyst sheets.

[1] Department of Applied Chemistry, Chuo University, Tokyo, Japan. [2] Research Institute, TOTO Ltd., Kanagawa, Japan. [3] Japan Technological Research Association of Artificial Photosynthetic Chemical Process (ARPChem), Tokyo, Japan. [4] Research Initiative for Supra-Materials, Interdisciplinary Cluster for Cutting Edge Research, Shinshu University, Nagano, Japan. [5] Office of University Professors, The University of Tokyo, Tokyo, Japan. ✉email: kkata@kc.chuo-u.ac.jp

Water splitting into hydrogen and oxygen by sunlight is one of the most promising clean energy resources. Tremendous efforts have been made mostly for the development of materials[1–3]. These have been extended for device fabrication and large-scale development[4,5]. In overall water splitting, careful selection of the band positions is necessary both for the reduction of the proton and the oxidation of water, for which co-catalysts are necessary to promote the reactions. Materials have emerged one after another, and one of the materials which can split water using a single semiconductor photocatalyst is aluminum-doped strontium titanate (SrTiO$_3$:Al). An apparent quantum efficiency has reached almost 100% for overall water-splitting under UV light illumination[6]; where the mechanism was studied[7]. Alternatively, two different materials have been used in combination, each for hydrogen and oxygen generation, where the remaining charges are compensated by charge transfer between the two materials, called "Z-scheme water-splitting system"[1,2,8–10].

In the Z-scheme strategy, various combinations of two different semiconductors were introduced. One of the favored combinations is BiVO$_4$ and Rh-doped SrTiO$_3$ (SrTiO$_3$:Rh) as oxygen and hydrogen generation catalysts. In the earlier stage of the Z-scheme researches, the photo-excited electrons in BiVO$_4$ were transferred to SrTiO$_3$:Rh via a redox shuttle in an aqueous solution. Still, the number of candidates for the redox shuttle is limited, such as IO$_3^-$/I$^-$, Fe$^{2+}$/Fe$^{3+}$. Next, the Z-scheme was demonstrated without using the redox mediator via direct contact between BiVO$_4$ and SrTiO$_3$:Rh semiconductors[11] where the photo-excited electrons and holes in each photocatalyst recombine directly at the interface. The conductive binders were searched and introduced to promote charge transport between two materials. Alternatively, particulate photocatalyst sheets in which the photocatalyst particles are fixed onto a glass substrate have also been reported. For example, a physical vapor deposition-based photocatalyst sheet composed of SrTiO$_3$:Rh,La and BiVO$_4$:Mo embedded on an evaporated gold layer can split water under the solar light with a solar-to-hydrogen conversion efficiency of 1.1%[12,13]. We also recently developed a photocatalyst sheet named a printable photocatalyst sheet that can be prepared via a facile and extensive screen-printing method incorporating a conductive colloidal binder (e.g., Au, ITO) with a highly packed film structure[13–15]. However, the STH was merely 0.4%[15], and it is necessary to understand the electron transfer process and design the optimal film structure for achieving STH>1%.

In water-splitting with photocatalysts, it is essential for photo-excited charge carriers to be separated and utilized for water oxidation and reduction without losing them due to recombination. Thus, charge carrier dynamics in various photocatalyst particles and films including SrTiO$_3$ or BiVO$_4$ have been studied. Transient absorption (TA) and time-resolved photoluminescence have been frequently utilized to understand the processes. Concerning SrTiO$_3$, various doping cause trap sites, and the lifetime of the charge carriers was extended from microseconds to milliseconds, and the effect was studied in relevance to the photocatalytic activity. In several reports, doping of a single element induced a recombination center and reduced the charge carriers, but co-doping could suppress the recombination[16–18]. The charge carrier dynamics were also studied for BiVO$_4$ on an ultrafast and wide time range. The effects of trap states were discussed mostly for elongation of hole lifetimes[19–22]. The effect of active and inactive oxygen defects was clarified[23]. A heterojunction of BiVO$_4$ with WO$_3$ was studied and revealed that the junction effectively removed electrons at the interband states[24] providing a benefit for water oxidation on a millisecond to second order[25].

Our approach to studying the charge carrier dynamics uses the special combination of the measurement of the refractive index change instead of the absorption change or photoluminescence[26,27] and its accompanying analysis method called the spectral clustering method. In the former measurement approach, monitoring the refractive index changes has merits in terms of the detection of the interfacial charge transfer, and also the limitation of the detection wavelength is relaxed due to the broad wavelength response. Thus far, this method has been applied for studying charge transfer at the interfaces of the photocatalyts[28] and dye-sensitized solar cells[29]. For detecting the refractive index change, we used the transient grating (TG) method and studied the dynamics of the charge carriers for TiO$_2$[30–32] and hematite photoanodes[33–35].

The charge carrier behavior depends on the local structure and is inhomogeneous in nature for the photocatalysts, typically composed of calcinated particles and aggregates. Many studies have been devoted to studying the spatio-temporal behavior of charge carrier dynamics using TA and photoluminescence microscopy on micro-scales[36–38] as well as the photocurrent behavior via microscopic photo-electrochemical measurements[39,40]. We also have extended our measurements of the refractive index change to the local mapping of the transient responses of photo-excited charge carriers (pattern-illumination phase microscopy (PI-PM)). By illuminating a pattern of light, the sequence of images due to the refractive index change was obtained, and the image quality was recovered by the image-recovery calculation techniques such as the principal component analyses and the least absolute shrinkage and selection operators in the frequency domain for noise reduction, incorporated from the informatics theory[41]. The lifetime distribution of the charge carriers for a TiO$_2$ particulate film was obtained, and the research clarified a broad range of the lifetime of charge carriers[42]. Furthermore, the local responses of charge carriers were categorized via spectral clustering, and the hidden local responses of the non-radiative exciton relaxations were found for higher pump intensities[43]. Thus, using the combination of PI-PM and the clustering method, we can investigate the position- and structure-dependent charge carrier behavior under light irradiation, and also assign the different types of carriers at specific positions.

These findings motivate us to apply this combination of PI-PM and clustering analysis methods for one of the most promising Z-scheme water-splitting materials (BiVO$_4$:Mo/SrTiO$_3$:Rh with a conductive colloidal binder (ITO)), which is a printable photocatalyst sheet. We could visualize the spatially resolved photocatalytic activity via the categorization of the charge carrier behaviors. This methodology could detect the real active and inactive sites in the photocatalytic device and will support the optimization of the active structure of the photocatalyst.

## Results and discussion

**Measurement of time-resolved image sequence.** The time-resolved experiments were performed on a visible-light responsive photocatalyst sheet composed of SrTiO$_3$:Rh, BiVO$_4$:Mo, and ITO (STOR/ITO/BVOM) in two different solvents (acetonitrile (ACN) and water). Figure 1 shows the image sequences of the refractive index change for STOR/ITO/BVOM in different solvents ((a) in ACN, (b) in water) observed by the PI-PM method. The measured region for (a) and (b) were exactly the same and just replaced the solvents. The image sequence by illumination of the UV pump pulse (wavelength: 355 nm, pulse width: 3–5 ns) was recorded by the green probe pulses (wavelength: 532 nm, pulse width: 3–5 ns). The optical resolution of this microscope was ~1 µm. The refractive index change is proportional to the number of excited carriers if the probe wavelength matches an optical transition; otherwise, it is proportional to the number of free carriers[44]. The region irradiated via the pump light corresponds to the brighter regions in the images. The contrast of images originating from the refractive index changes due to

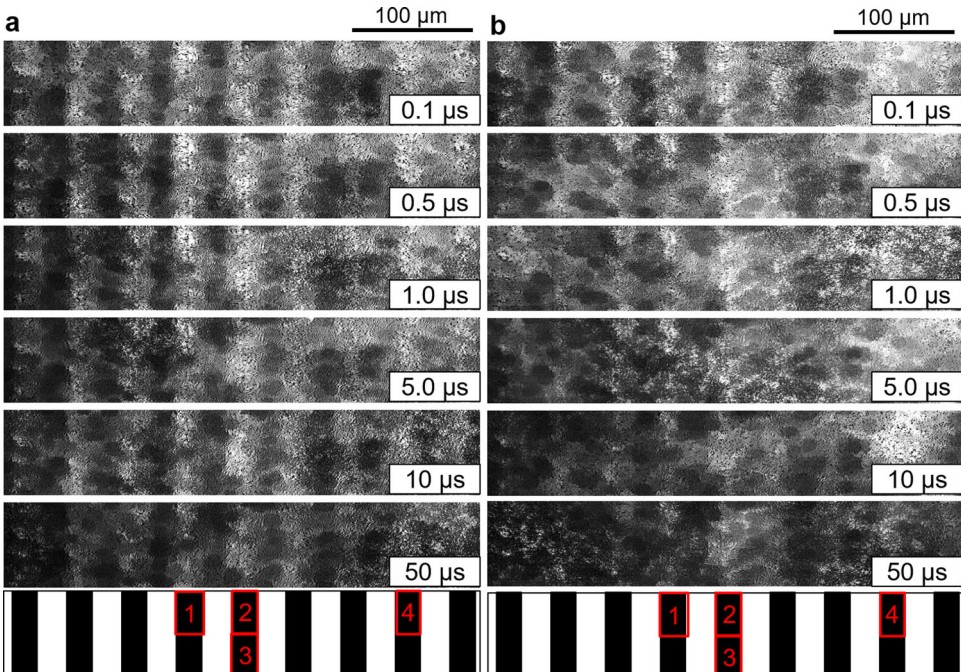

**Fig. 1 The image sequences of the patterned-illumination time-resolved phase microscope (PI-PM) in different solvents.** The images correspond to the refractive index changes after photo-excitation by a UV pump light for SrTiO$_3$:Rh/ITO/BiVO$_4$:Mo (STOR/ITO/BVOM) Z-scheme water-splitting sheet measured by the PI-PM method (**a** in ACN, **b** in water). The light intensity patterns of the pump light are shown at the bottom of each image sequence, and the selected regions for the cluster analysis are indicated in red squares. The result of the cluster analysis for area No. 1 is shown in Figs. 4–6, and the others are presented in Supplementary Figs. 4–8).

photo-excited charge carriers in these regions. Although the images in ACN retained overall brightness until 10–100 μs, the bright regions in water gradually faded out for 1 μs. The overall refractive index change decayed due to charge recombination; however, images in ACN did not show a simple single-component decay. Furthermore, it was noticed that the brightness in the excited regions in the images strongly depended on the sample position meaning that the materials in this photocatalyst sheet were not homogeneously distributed on the substrate.

**Temporal responses and assignments.** Temporal changes in the refractive index images were obtained to make an assignment of the responses for the STOR/ITO/BVOM photocatalyst sheet in two different solutions. These were calculated by taking the Fourier amplitude of the periodic stripe pattern at a spatial frequency corresponding to the inverse of the stripe spacing (Fig. 2a). A detailed calculation process was written in the previous papers[41,42]. The signal response for the STOR/ITO/BVOM in ACN was categorized into three response components: (1) rapid increases for 30 ns and then a decay for 100 ns (first response), (2) an additional rise for 1 μs, and decay for a couple of μs (second response), and (3) a plateau until 10 μs and decay for 100 μs (third response).

Figure 3 shows the scheme that photo-excited charge carriers were generated in the conduction band and valence band after photo-excitation. It is noted that the wavelength of the pump laser was 355 nm, and its energy exceeds the bandgap energy of both materials (BiVO$_4$ = 2.4 eV, SrTiO$_3$ = 3.2 eV). Therefore, we monitored the charge carrier dynamics after the interband transition of both materials. These charge carriers were rapidly trapped in shallow or deep trap states originating from multiple types of defects. This trapping process almost finishes within the picosecond time region[21,22,45,46]. These trapped charge carriers decayed due to the recombination inside the materials or the

extraction outside the materials. The first component is a mixture of these fast charge carrier dynamics. ACN is an inert solvent for charge carriers, and the carriers were confined inside the materials. Thus, these carriers remained on a slower time scale, and the response was observed for hundreds of microseconds, which corresponds to the microsecond recombination for BiVO$_4$ and SrTiO$_3$[16,25] included in the third component.

On the other hand, the second component in ACN disappeared in the water, as shown in Fig. 2a. It indicates that the second component (100 ns to 1 μs) in ACN was attributed to electrons in STOR or holes in BVOM on the surfaces because these surface-trapped charge carriers were consumed by water and utilized for water-splitting reactions. The loss of the component by water indicates that the corresponding charge carriers were used for water-splitting reactions, which were ensured by the fact that this photocatalytic sheet had a high water-splitting efficiency even for pure water (Supplementary Fig. 11). Vise versa, this result demonstrated that the surface-trapped carriers were effectively extracted to water in this system.

The working mechanism of the Z-scheme water-splitting system is complicated, and many processes of photo-excited charge carriers are included in the response. To simplify the assignment, we separately investigated the charge carrier dynamics of BVOM and STOR using the same materials used for the Z-scheme system. Figure 2b, c show the refractive index responses obtained for BVOM and STOR in ACN and water, respectively. We could confirm that the second components (100 ns to 1 μs) were observed in ACN in both samples, and they disappeared in water in both cases. These results supported that these second components were attributed to the holes in BVOM and electrons in STOR, respectively, and they were utilized for water-splitting reactions, as shown in Fig. 2a. We supposed that the holes in BVOM and electrons in STOR for water splitting were observed as a mixture in the response of the Z-scheme sample because these lifetimes were matched by chance. In

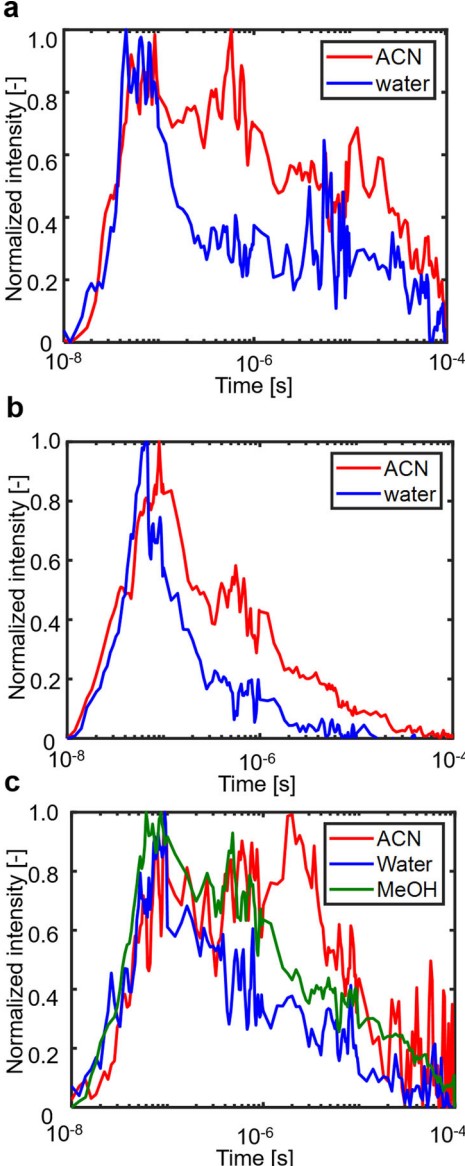

**Fig. 2 Average refractive index change responses. a** SrTiO₃:Rh/ITO/
BiVO₄:Mo, **b** BiVO₄:Mo only, and **c** SrTiO₃:Rh only until 100 μs in
acetonitrile (ACN) and water obtained from the image sequence obtained
by the patterned-illumination time-resolved phase microscope. (red: in
ACN, blue: in water) The response was obtained from the signal amplitude
at the light irradiated regions. The response in methanol (MeOH) is also
shown for **c** STOR only (green: in MeOH).

addition to this finding, we recognized that the second rising
component was delayed until 10 μs in STOR in ACN. STOR has a
mid-gap state corresponding to the $Rh^{3+/4+}$ state within the
bandgap. Rh is doped for the visible-light absorption via
the transition from the $Rh^{3+}$ state to the conduction band
and the valence band to the $Rh^{4+}$ state corresponding to 2.3 and
2.7 eV, respectively[18,47]. The considerable delay of the response in
STOR implies that the response of STOR in ACN include unique
components of photo-excited charge carriers due to the loss of the
water-splitting reactions in ACN instead of water.

We assumed that the photo-excited holes at the valence band
in STOR decayed to the $Rh^{3+}$ state observed as the slow-rising
component (formation of $Rh^{4+}$ state) until 10 μs in ACN. Since it
was reported that holes in the $Rh^{3+}$ states could be scavenged by

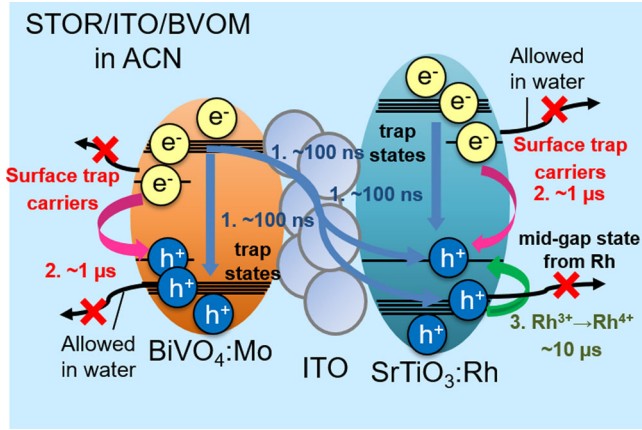

**Fig. 3 Schematic drawing of the entire charge dynamics.** This scheme
corresponds to the charge transfer diagram in SrTiO₃:Rh/ITO/BiVO₄:Mo
(STOR/ITO/BVOM) in acetonitrile (ACN). Step 1 includes intrinsic
recombination inside materials and charge transfer between two materials.
Step 2 indicates the decay of surface-trapped charge carriers in ACN, which
are utilized for water-splitting reactions with water outside. Step 3 includes
the hole trapping to $Rh^{3+/4+}$ state in STOR and the slower recombination.
Surface-trapped carriers cannot be transferred to the solution side in ACN,
where all processes of charge dynamics are completed in the system.

MeOH, we measured the response of STOR in MeOH. The result
is shown in Fig. 2c, and it clearly demonstrated that the slow-
rising component in ACN was quenched in MeOH. Thus, we
concluded that this rising component was attributed to the hole
decay process to $Rh^{3+}$ state, corresponding to the oxidation
process from $Rh^{3+}$ to $Rh^{4+}$ states in STOR.

**Charge transfer scheme and its kinetics.** A general under-
standing of the mechanism of the Z-scheme system is explained
here based on the description provided by Osterloh et al.[48]. Upon
irradiation with visible light, both BVOM and STOR are photo-
excited, and the photo-generated holes in BVOM and photo-
generated electrons in STOR are used for water splitting to
oxygen and hydrogen, respectively. Photo-excited electrons in
BVOM are transferred to the neighboring STOR particles, and
the $Rh^{4+}$ are reduced to $Rh^{3+}$ via electron transfer. The $Rh^{4+}$
works as the electron acceptor in this case. With regard to STOR,
electrons are excited from the $Rh^{3+}$ state to the conduction band
for the visible-light absorption causing the generation of $Rh^{4+}$
species. Water splitting continues when these reactions and
charge transfer cycles repeatedly occur. The $Rh^{4+}$ states reduce
the activity of water splitting by working as a recombination site
for the electrons in the conduction band, and much effort has
been made to reduce the $Rh^{4+}$ states[18,47].

The schematic flow of the whole charge dynamics of STOR/
ITO/BVOM is summarized in Fig. 3. Based on the series of results,
we concluded that the response included at least three
components. As a first step, photo-excited charge carriers were
decayed for <100 ns due to the intrinsic recombination processes
because the responses in this region were not influenced by water.
During these processes, free charge carriers and trapped charge
carriers at shallow trap states recombined inside each material
(first component). The processes could include the recombination
between the electrons in BVOM and the holes in STOR as well
because there was smooth charge transfer via the ITO mediator. In
the second component, the response decayed due to the
recombination of the surface-trapped charge carriers in ACN
until ~1 μs. The components were lost for water splitting when the
photocatalytic sheets contacted water. This response includes the

contributions of both the electrons in STOR and the holes in BVOM considering the responses of STOR and BVOM only (Fig. 2b, c). The energy levels for these trap states have been not fully clarified, and there were reports that the energy levels 0.3 eV lower than the conduction band for STO[49] and 0.1 eV above the valence band[50] for BVO could be used for water-splitting. The third component until ~100 μs includes the hole accumulation, which decayed from the valence band to $Rh^{3+}$ state in STOR (~10 μs) because this component was observed for STOR only and was quenched by MeOH (Fig. 2c). The third component also includes other slower recombination processes. The hole accumulation in STOR contributed less to the Z-scheme system because ITO worked to mediate the electron transfer in BVOM to STOR and effectively reduced the $Rh^{4+}$ state. However, we confirmed that the accumulation of $Rh^{4+}$ (slow-rising component) was enhanced due to inefficient electron transfer from BVOM without the charge transfer mediator (ITO mediator) (Refer to Section of "The effect of ITO mediator" and Supplementary Figs. 1 and 2 for a detailed discussion on the ITO mediator).

In terms of the water splitting, the effective charges on each material in the second process are essential, and the high efficiency of water splitting of this Z-scheme system is possibly due to the well-matched lifetimes of the surface-trapped electrons in STOR and holes in BVOM. Furthermore, the oxidized $Rh^{4+}$ state was effectively reduced by the smooth electron transfer from BVOM; otherwise, the state is well-known to deteriorate the hydrogen-generating electrons in STOR due to the recombination.

The kinetic analysis for the Z-scheme system is provided in "Phenomenological kinetic analyses for charge carrier dynamics in Z-scheme materials" in Supplementary Information. The phenomenological analysis of the rate equations is described in a scheme, (Supplementary Fig. 12) and the charge carrier response with a similar temporal response of the refractive index change was reproduced. Furthermore, the reduction of the Rh state density was demonstrated by increasing the charge transfer rate due to the charge mediator. (Supplementary Fig. 13). The final responses of the refractive index change consist of multiple charge carrier responses, but reliable combination of kinetic parameters was not obtained due to multiple choices and the low SNR of the responses.

**Clustering analysis and mapping**. Many types of charge carrier responses were overlapped for the Z-scheme systems because they consisted of more than one material. Furthermore, we noticed the inhomogeneity of the responses in the image sequences measured by the PI-PM method. Hence, we categorized the position-dependent (local) responses based on their similarity via clustering analysis of the transient image sequence. A detailed procedure of this analysis was introduced in our previous paper[43]. In brief, one of the local regions in the excited area was selected for the analysis, and the local responses in the selected region were extracted from each pixel in the image sequence. These responses were classified into several categories of the charge carrier responses based on the similarity calculated by the rise and decay profile. We applied this analysis to the local responses in the image sequence of STOR/ITO/BVOM in ACN (Fig. 4). The selected position for the analysis is indicated in the red square in Fig. 1, bottom (No. 1). Figure 4a shows a sequence of the temporal images of the refractive index change representing a position-dependent response where some micron-sized particles showed a large refractive index change with higher brightness lasting until 1 μs and fading out for 10 μs.

The transient responses were classified into three categories from the spectral clustering[51] and were mapped in Fig. 4b. The

outliers (far from the three categories) were indicated as #0 (black) in this mapping. The original microscopic phase image of the sample was measured by the same optical setup as the PI-PM method without the irradiation of the pump pulse and is shown in Fig. 4c. In the sample image, micron-sized aggregates were seen. The bright positions lasting on the order of microseconds in the transient image sequence were frequently overlapped with the positions of the aggregations. The aggregates correspond to STOR, which could be confirmed by an SEM (Supplementary Fig. 3a) and the corresponding energy-dispersive spectroscopy (EDS) result of element Sr in Supplementary Fig. 3b. The synthesized STOR particles with a typical diameter of hundreds of nanometers easily form aggregates while micron-sized BVOM particles were well-dispersed recognized by element Bi (Supplementary Fig. 3c). Some aggregates of STOR remained even after mixing the particles to form a Z-scheme sample.

In each category, the responses at all pixels in the same category were averaged and shown in Fig. 4d. The categorized map in Fig. 4b, d was mostly composed of red and blue regions. Based on the assignment in Fig. 2a, the response in the red region was similar to an averaged overall response (data 1 in d), and the blue response seems to be composed of the first and third responses (data 2 in d). There was a small portion of green regions in Fig. 4b, and it showed a slow-rising component corresponding to the third component (data 3 in d). Based on the assignment described earlier, the response of data 1 in Fig. 4d was composed of the intrinsic recombination and the decay of the surface-trapped carriers until 1 μs, utilized for water-splitting reactions. Therefore, water-splitting reactions should mostly occur in the red region. The blue response was composed of the intrinsic recombination. The area of the green regions corresponding to the hole decay to $Rh^{3+}$ state ($Rh^{4+}$ formation) in Fig. 4d was much smaller compared with the red and blue regions. This means that the blue and green regions do not have a water-splitting activity.

This analysis was applied to other regions (No. 2 to No. 4 in Fig. 1c), and the results are shown in Supplementary Figs. 4–6. Supplementary Figures 4 and 5 had almost the same categories of the responses as Fig. 5. However, the red regions mostly occupied the whole area compared with Fig. 5. This means that regions No. 2 and No. 3 were more active for water-splitting reactions than that for region No. 1. On the other hand, as shown in Supplementary Fig. 6, the green region increased for region No. 4. This indicated that the $Rh^{4+}$ formation process was observed over a large portion of this region leading to ineffective water-splitting. Overall, we often observed the slow responses at the aggregates of STOR, but the regions did not always match their positions. The optical image could provide aggregates on the surface, and the aggregates could possibly be buried in the film. Based on these results, each material was not dispersed homogeneously and mixed well on the substrate indicating that some regions were not ideal for STOR/ITO/BVOM compositions.

**Effect of mediator and water**. The same analysis was conducted in the case of a Z-scheme sample without the ITO mediator (STOR/BVOM) to confirm the effect of ITO (Fig. 5). The results show that the same types of categories for transient responses were found. However, the area of the green regions in the categorized map (Fig. 5b) drastically increased while the number of the red regions decreased considerably compared with Fig. 4b. The ratio of the blue region remained almost the same. This result indicated that the area of the $Rh^{4+}$ formation drastically increased if there was no ITO mediator in the system. This is exactly the reason why the water-splitting reaction was inefficient for this system. The same experiments and analyses were applied

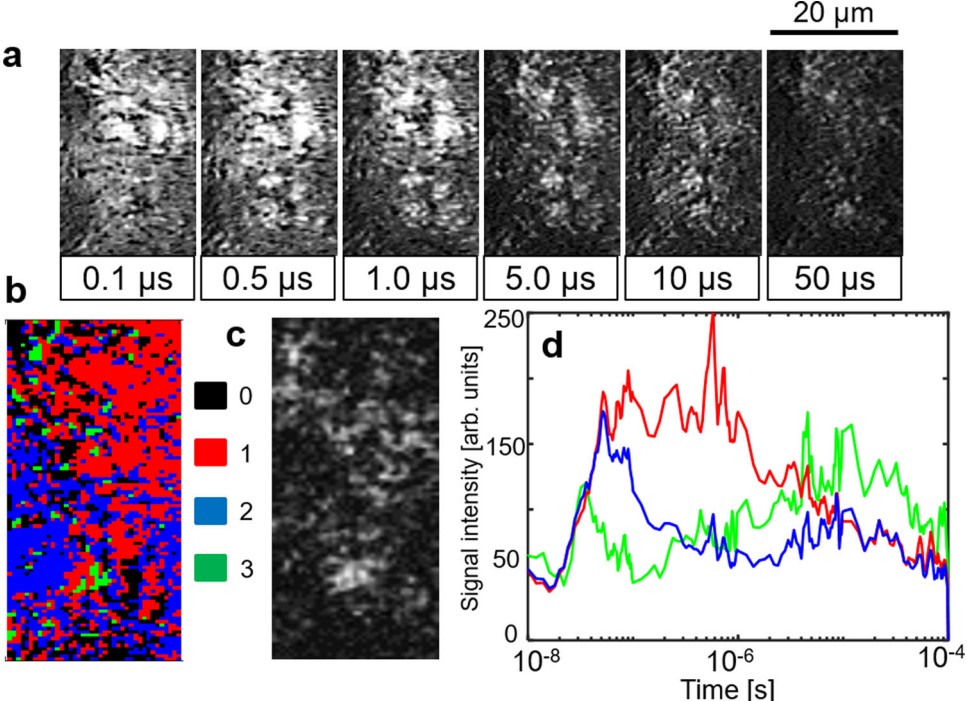

**Fig. 4 Image sequence and clustering analysis with mediator. a** An image sequence of the refractive index change for SrTiO$_3$:Rh/ITO/BiVO$_4$:Mo (STOR/ITO/BVOM) in acetonitrile (ACN) in a square region (20 × 50 µm) corresponding to no. 1 in Fig. 1, bottom on the order from nanoseconds to microseconds. The scale bar corresponds to 20 µm. **b** The categorized mapping of the charge carrier responses of **a**. An outlier positioned far from all categories was colored in black (#0). **c** A microscopic image in the same area as in **a**. **d** The averaged responses for each category in **b** are shown. (red: category 1, blue: category 2, green: category 3).

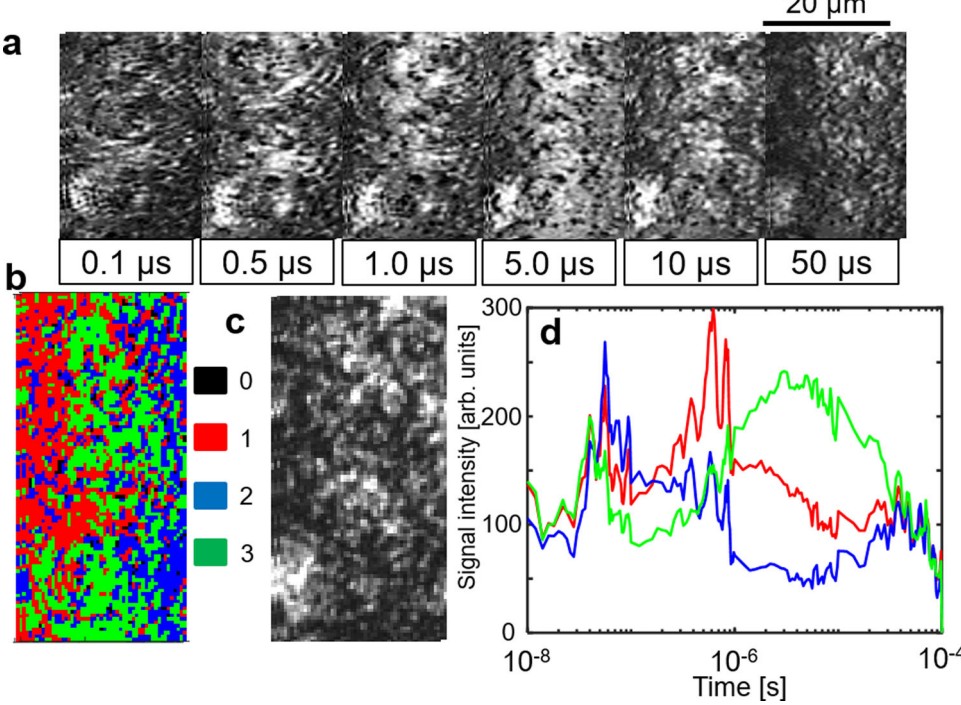

**Fig. 5 Image sequence and clustering analysis without mediator. a** An image sequence of the refractive index response for SrTiO$_3$:Rh/BiVO$_4$:Mo STOR/BVOM in acetonitrile (ACN) in a square region (18 × 50 µm) corresponding to no.1 in Fig. 1c on the order from nanoseconds to microseconds. The scale bar corresponds to 20 µm. **b** The categorized mapping of the charge carrier responses of **a**. An outlier positioned far from all categories was colored in black (#0). **c** A microscopic image in the same area as **a**. **d** The averaged responses for each category in **b** are shown. (red: category 1, blue: category 2, green: category 3).

to other regions (No. 2 and No. 3 in Fig. 1, bottom), and the results are shown in Supplementary Figs. 7 and 8. We confirmed that the $Rh^{4+}$ formation occurred in a large portion of these regions. There was a little correlation between the surface aggregates and the green regions in absence of the ITO mediator, and this indicates that the key process for the reduction of the $Rh^{4+}$ states is to promote the charge transfer between BVOM and STOR. From the combination of the transient image sequence and the spectral clustering, we could successfully visualize the position where the water-splitting reactions proceeded and clarify the role of ITO for higher efficiency. Since the positions for the high and low efficiencies of water-splitting reactions were spatially separated, this indicates that the charge transfer between BVOM and STOR was efficient in some regions and not in other regions and possibly could be improved by the mixing and deposition conditions on the substrate.

Finally, we studied that the responses of STOR/ITO/BVOM in water by using the same cluster analysis to compare the difference from the results in ACN (Fig. 6). The analysis was applied for exactly the same region as the region in Fig. 4, and only the solvent was replaced with water. Figure 6b, d show that the categorized map was composed of the blue and green regions corresponding to the fast decay (data 1) and the weak response (data 2) regions. We could not observe the red and green regions found under the ACN condition (Fig. 4). This result indicated that all surface-trapped charge carriers in the case of ACN were utilized for water-splitting reactions in water. These components were completely removed by water, and only intrinsic recombination was observed. A comparison of Figs. 4 and 6 ensures the reaction of water splitting on the surface.

The categorization of local charge carrier responses with a combination of the PI-PM measurements and the clustering analyses reported herein provides useful information for designing and fabricating highly efficient photocatalytic films or devices

that have heterojunctions between different semiconductor particles.

In summary, this work revealed the origin of the efficient water-splitting reactions in STOR/ITO/BVOM photocatalyst sheets that can split water to $H_2$ and $O_2$ under visible light by employing our original clustering analysis of photo-generated charge carrier responses in the transient image sequence. The combination of the observation technique (PI-PM) and the clustering analysis could visualize reactive sites for efficient water-splitting reactions and inactive sites deteriorating water-splitting efficiency. We found three components from the averaged responses of the refractive index change for different combinations of materials in various solvents corresponding to the intrinsic recombination of photo-excited charge carriers inside materials (~100 ns), recombination of surface-trapped charge carriers which can essentially contribute to split water (~1 μs), and the hole trapping (accumulation) to the Rh state in STOR generating an $Rh^{4+}$ state (~10 μs). Consequently, we visually demonstrated the region of hole trapping to the Rh state where the region suppresses the water-splitting reactions in the photocatalyst sheet. This suppression decreased in the sheet including ITO mediators at the reaction sites due to the efficient inter-particular charge transfer. This technique will be a powerful tool not only for the detection of active/inactive sites for the water-splitting materials but also for the other materials involving electron transfer reactions, providing swift optimization for photocatalytic materials.

## Methods

**Time-resolved imaging**. In the PI-PM method[41,43], an arbitrary light pattern is irradiated on a sample to excite charge carriers. The charge carriers decay or diffuse as time passes due to the recombination, charge trapping, and transport, and the charge carrier distribution varies in time. The distribution of the photo-excited charge carriers is observed from the refractive index change via phase-sensitive imaging. The refractive index was imaged by the Talbot self-imaging technique[52].

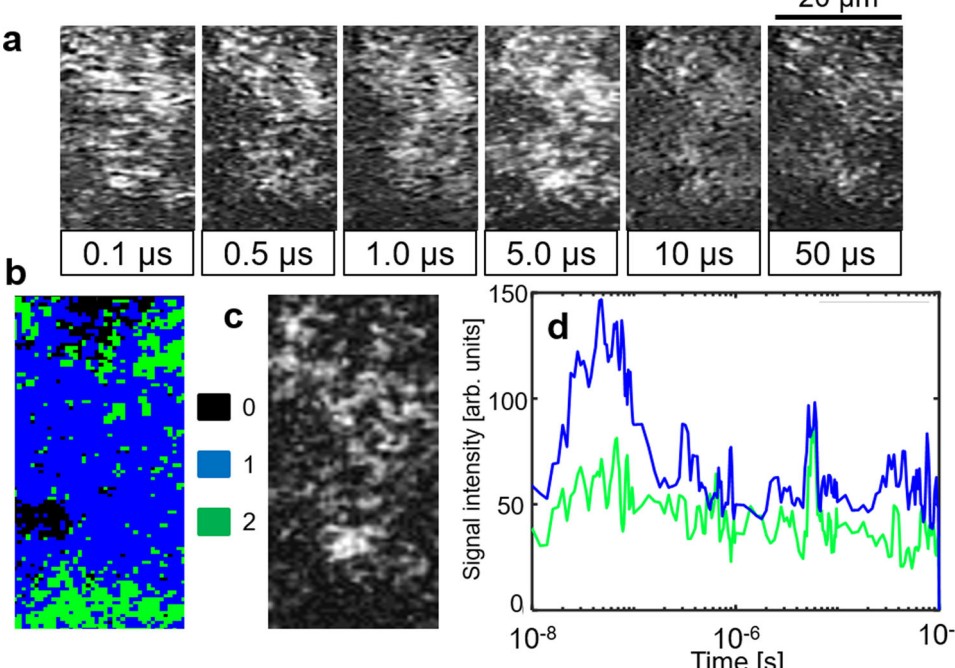

**Fig. 6 Image sequence and clustering analysis in water. a** An image sequence of the refractive index response for SrTiO₃:Rh/ITOBiVO₄:Mo (STOR/ITO/BVOM) in water in a square region (20 × 50 μm) corresponding to no. 1 on the order of nanoseconds to microseconds. The location of the sample was exactly the same as Fig. 5a. The scale bar corresponds to 20 μm. **b** The categorized mapping of the charge carrier responses of **a**. An outlier positioned far from all categories was colored in black (#0) **c** A microscopic image in the same area as **a**. **d** The averaged responses for each category in **b** are shown. (blue: category 1, green: category 2).

The pattern-illumination is required to apply various image-recovery algorithms[41]. In this study, only the background correction was processed before the analyses to prevent the loss of small features in images. In this optical configuration, the TA responses could also be included in the image, but the refractive index change was the major contribution to the signal, which could be confirmed by optimizing the focus position because the TA signal intensity does not depend on the focus position.

The experimental setup is fully described in Supplementary Fig. 9. In brief, the wavelengths of the pump and probe pulse lights were 355 and 532 nm, respectively, with pulse widths of 3–5 ns. The pump light was illuminated as a Ronchi ruling pattern similar to the transient grating technique reported previously to understand the refractive index changes via amplitude changes via Fourier transform. The width of each line in the ruling pattern was 25 μm, and the spacing was 45 μm.

**Preparation of photocatalysts**. The sample was Rh-doped $SrTiO_3$ (STOR) and Mo-doped $BiVO_4$ (BVOM) including ITO nanoparticles as a conductive binder. These were printed on a glass substrate with a thickness of 1 μm. The detailed preparation method is described in SI. The SEM images of STOR and BVOM are shown in Supplementary Fig. 10 in SI. The average diameters were 300 nm and 2 μm, respectively. The gas evolution data for this photocatalyst sheet in pure water are shown in Supplementary Fig. 11, and the STH was 0.4%. The SEM image of a photocatalytic sheet is shown in Supplementary Fig. 3, together with EDS analyses. The BVOM particles were well-dispersed, while STOR particles formed aggregates.

**Measurement cell**. The solid/liquid interface was prepared by putting another glass slide together with a silicon rubber spacer (thickness: 0.5 mm), and liquids were sandwiched within the gap. Each film sample was measured in contact with ACN and water. ACN is an inert solvent for photocatalytic reactions where no charge transfer at the interface is guaranteed[28]; this does occur in water. We used methanol (MeOH) as a hole scavenger for assignment and confirmation of the hole responses.

## Data availability

All the source data of the time responses in this paper, the time-resolved image sequences, the clustering analyses, and the scheme figures are available via https://doi.org/10.6084/m9.figshare.14673111.

## Code availability

The code used in this paper for the clustering analysis is available via https://doi.org/10.6084/m9.figshare.14673120.

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

## Acknowledgements
The research was financially supported by JST PRESTO (#JPMJPR1675), and the Institute of Science and Engineering, Chuo University.

## Author contributions
M.E., H.T., and K.K. designed the experiments and made measurements. T.I., S.O., H.T., and K.D. prepared the photocatalytic sheet. M.E. analyzed the data, and M.E., H.T., K.D., and K.K. analyzed the results. M.E. and K.K. wrote the manuscript, and all the authors reviewed it.

## Competing interests
The authors declare competing interests.
