## [Peer Review File · Nature Communications]

REVIEWER COMMENTS

Reviewer #1 (Remarks to the Author):

The manuscript "Charge Carrier Mapping for Z-scheme Photocatalytic Water Splitting Sheet by Categorization of Microscopic Time-resolved Image Sequence" by Ebihara et al. reports on interesting experimental insights in the function mechanism of a composite material not only with temporal but also with spatial resolution. The system investigated is a Z-scheme catalyst with proven overall water splitting capability. Pattern-illumination time-resolved phase microscopy is used to analyze changes in refractive index of the sample reporting on charge carriers generated upon excitation and their dynamics. The spatial resolution of the imaging technique allows the identification of the distribution of certain dynamic signatures in the catalyst material and relates it to regions of high and low performance respectively. The research is definitely interesting and of highest importance to support development of structures capable of efficient overall water splitting.

Although in my opinion high potential of this research there are a number of issues which in my opinion need to be addressed, before this work can be regarded ready for publication. Besides the research itself, the presentation is poor and the manuscript needs major revision in this respect. A thorough check of language is highly recommended. A list of specific issues which needs to be addressed you will find below. In my opinion the manuscript is not suited for publication in Nature Communications in the current form. After major revision, the manuscript could be considered again.

- 1) What is the spatial resolution which can be reached with the imaging technique? Could the authors comment on this?
- 2) Please define the excitation conditions and probing conditions more clearly in the text (it is given in the methodology, this is ok, but it would be very convenient for the reader to give this either in the text, e.g. first paragraph of the results and discussion part, once or in the caption of the figures).
- 3) Is the data collected in water and acetonitrile shown in Figure 1 from the same sample and from exactly the same regions? Please comment on that. I assume, this is the case. But if not, how can you make sure the effect you observe are not a matter of imaging different regions of the sample?
- 4) Figure 1 and also the other images: The black/white contrast is very poor in the images, and after printing probably not well visible anymore. Is there a way to enhance the quality? Further, the illumination pattern would be nice to have directly below the data displayed in the same scale. This makes the localization of the regions under investigation much easier. Further, if I read the figure correctly region 4 is hidden under the legend in the single images. Please shift the legend to a region which is not investigated further or find a different representation of the time scale without hiding parts of the images.
- 5) I recommend avoiding the use of red/green contrast in the graphs, which might challenge parts of the potential readers.
- 6) Figure 2/3/4: Here I suggest a different representation which also offers better comparability between the data collected for the complete Z scheme system and the single components. Further it will allow to avoid showing the action scheme twice. I suggest to prepare one figure containing all the data for direct comparison next to each other (i.e. Figure 2 panel a and Figure 3) and have another figure just containing Figure 4 and deleting Figure 2 panel b, which is not necessary.
- 7) Could the authors shift the information on the hole quenching by methanol as proof for the nature of the second response in STOR arising from hole states mentioned on page 8 to the description of the data and assignment of the features on page 7? In my opinion this supports the flow of the logic in the presentation and discussion of the data. The data collected for STOR in presence of MeOH could even be added to Figure 3 panel b. This could nicely illustrate the point that the slower response 2 in STOR which is not present in the complete system is related to unquenched holes.
- 8) The authors investigate the local responses and draw conclusions on the local activity of their sample. Are there microscopy images of the same areas of the sample so the spatial distribution derived from cluster analysis could be directly correlated with structural features visible under the microscope? Are there any inhomogeneities visible which could be directly related to the distribution of responses mapped out from the PI-PM investigation? I think in Figure 5c, something in this direction is

shown, please specify which microscopy technique this image is resulting from. In the text is mentioned the "original microscopic phase image", I assume this means before excitation? Are there further images collected by other techniques which could give additional information on heterogeneity and distributions of defects in the printed system, etc. which could be related to the signals collected? The authors relate the structure which can be recognized in Figure 5c to aggregations. Do you think these are aggregation of all three materials, leading to active regions, and in the other regions the material is not properly assembled and hence shows no activity? Is there a way to show this by imaging the sample composition with some chemical contrast? Also considering the images of the other investigated regions I think there the correlation between aggregations observed in the original phase image and the signatures from the analysis of the time resolved data is hard to see. From my point of view a convincing assignment, whether these aggregates are somehow related to a certain feature in the dynamics based on the presented data is not possible.

9) SEM images are provided of the single components but not of the printed system. This would be interesting to get an impression on the heterogeneity of the water splitting sheet.

10) The authors relate the feature in the dynamics which is only present in ACN to the active regions of the sample. I agree, there are certain arguments for that. I think it would be interesting to use spatially resolved electrochemical microscopic techniques, e.g. scanning electrochemical probe microscopy (SEPM) to map H₂ evolution, on such a material to be able to actually correlate active and non-active regions with the results from the PI-PM investigation. This would give a real correlation between the observed spatial distribution of features in the dynamics and activity of the sample.

Grammar issues to correct (list not complete)

11) Abstract last sentence: "This combination methodology could not only lead to the maximum performance of photocatalyst sheets but also applicable to other 28 systems involving electron transfer." Something is missing here, please check this sentence.

12) Page 3 line 90/91: "...local mapping of the transient responses of photo-excited charge carriers. (pattern- illumination phase microscopy (PI-PM)). "

Reviewer #2 (Remarks to the Author):

The authors studied charge carrier dynamics of Mo-doped BiVO₄ and Rh-doped SrTiO₃ samples in acetonitrile and water via a novel PI-PM technique. The method can spatially resolve carrier dynamics. By studying the materials in isolation and in a heterostructure in two different solvents, the authors were able to assign features in the PI-PM dynamic trace to carrier dynamic pathways. The work focuses on state-of-the-art materials and uses a novel, exciting, and interesting technique to learn about the underlying fundamental photocatalytic processes related to solar water splitting. Hence, the nature and scope of the work is appropriate for Nature Communications. I think many researchers in the field would find this study interesting. The data generally supports the authors' conclusions, but the manuscript requires two major revisions before publication. First, the authors qualitatively discuss the time scales and processes. Second, the authors should expand a bit on their discussion of how PI-PM works generally so readers who are not familiar with the previous 2 papers can understand the significance of the results. Hence, I recommend the paper be published after a major revision.

Major comments:

1. The impact of the work would be improved if the authors could develop a model to fit the time-dependent PI-PM data. This model would extract rate constants and provide deeper physical insight to the systems. Several models have been developed to quantitatively analyze time- or frequency-resolved data in the field of photoelectrochemistry (Peter, L. M.; Vanmaekelbergh, D. Time and Frequency Resolved Studies of Photoelectrochemical Kinetics. In *Advances in Electrochemical Science and Engineering*; John Wiley & Sons, Ltd, 1999; pp 77–163. <https://doi.org/10.1002/9783527616800.ch2>). Since this is the third paper on the PI-PM technique, it seems appropriate that the authors would quantitatively extract information from the data.

2. The imaging capability of the PI-PM method is not clear because the PI-PM image is not compared to the region of interest in a side-by-side fashion. The reader has no idea if the signal is due to the noise or the presence of larger/smaller particle. The authors should correlate the PI-PM images with SEM images of the same region of the sample.

3. Figure 5-7 are essentially the same measurements on different samples and/or under different conditions. If the authors could quantitatively analyze the raw data and extract information, it would allow them to construct new knowledge, which could lead to new figures instead of repetitive figures.

4. When PI-PM is introduced in line 113, page 4, it would be good if the authors could explain the physical reason why the refractive index changes and where it occurs (e.g., in the solid, in the electrolyte, or at the interface). How is the refractive index change quantitatively linked to the total number of charge carriers in the semiconductor? More explanation of the PI-PM technique in the main text would be useful here.

5. For all the PI-PM images, can the authors show the SEM image of the same area of the sample? The images in Figure 5-7 are not very helpful.

6. What is the fundamental spatial resolution of PI-PM versus what is being achieved in this study?

Minor comments:

1. Page 3, line 92: what does "the image quality was recovered by the image" mean, exactly?

2. Is "ITO binder" in line 281 the best term?

3. Line 311... define "local".

Reviewer #3 (Remarks to the Author):

In "Charge Carrier Mapping for Z-scheme Photocatalytic Water Splitting Sheet by Categorization of Microscopic Time-resolved Image Sequence", Ebihara and co-workers report on an interesting method to visualize charge transfer on particles during water plaiting. The paper is well written and the provided data sufficient. Some comments are provided below:

- On page 7 I could read "it is supposed that the holes in BVOM and electrons in STOR for water splitting were observed as a mixture in the response of the Z-scheme sample because these lifetimes matched by chance." Would it be possible to differentiate the process in one or the or the particle using laser excitations of various wavelength where only one of the two material is generating photo carriers?

- Information about the origin of the energy levels responsible for trapping in "Step 2" should be included if known.

- Could the charge transfer mapping (e.g. Fig 5.b) be assigned to any specific morphology or micro-structural defect in the particle?

Reviewer 1 comments

Reviewer: 1

Comments:

The manuscript “Charge Carrier Mapping for Z-scheme Photocatalytic Water Splitting Sheet by Categorization of Microscopic Time-resolved Image Sequence” by Ebihara et al. reports on interesting experimental insights in the function mechanism of a composite material not only with temporal but also with spatial resolution. The system investigated is a Z-scheme catalyst with proven overall water splitting capability. Pattern-illumination time-resolved phase microscopy is used to analyze changes in refractive index of the sample reporting on charge carriers generated upon excitation and their dynamics. The spatial resolution of the imaging technique allows the identification of the distribution of certain dynamic signatures in the catalyst material and relates it to regions of high and low performance respectively. The research is definitely interesting and of highest importance to support development of structures capable of efficient overall water splitting.

Although in my opinion high potential of this research there are a number of issues which in my opinion need to be addressed, before this work can be regarded ready for publication. Besides the research itself, the presentation is poor and the manuscript needs major revision in this respect. A thorough check of language is highly recommended. A list of specific issues which needs to be addressed you will find below. In my opinion the manuscript is not suited for publication in Nature Communications in the current form. After major revision, the manuscript could be considered again.

Author reply:

We would like to appreciate your evaluation of our paper and very happy to hear it. As you mentioned, there were several unclear presentations and language usage. We have revised the presentations according to your advice and had a check on the language throughout the manuscript by a commercial editing service. We hope these revisions would be satisfactory for you.

- 1) What is the spatial resolution which can be reached with the imaging technique? Could the authors comment on this?

Author reply:

The optical resolution of images was $\sim 1 \mu\text{m}$. From the observation, a spot with a diameter of 500 nm could be recognized, but we could safely say that the resolution is less than $1 \mu\text{m}$ because sometimes we could observe a ring pattern for a small object, which is due to the diffraction. Only the resolution was described in the sentence.

(p. 4 line 16)

- 2) Please define the excitation conditions and probing conditions more clearly in the text (it is given in the methodology, this is ok, but it would be very convenient for the reader to give this either in the text, e.g. first paragraph of the results and discussion part, once or in the caption of the figures).

Author reply:

The pump and probe light conditions (wavelength and pulse width) are described at the initial part of the results and discussions.

(p. 4 line 12-14)

- 3) Is the data collected in water and acetonitrile shown in Figure 1 from the same sample and from exactly the same regions? Please comment on that. I assume, this is the case. But if not, how can you make sure the effect you observe are not a matter of imaging different regions of the sample?

Author reply:

We are sorry for this confusion. The measured regions were totally the same for the data in Figure 1 in water and acetonitrile. The sentence has been modified to indicate it.

(p. 4 line 11-12)

- 4) Figure 1 and also the other images: The black/white contrast is very poor in the images, and after printing probably not well visible anymore. Is there a way to enhance the quality? Further, the illumination pattern would be nice to have directly below the data displayed in the same scale. This makes the localization of the regions under investigation much easier.

Further, if I read the figure correctly region 4 is hidden under the legend in the single images. Please shift the legend to a region which is not investigated further or find a different representation of the time scale without hiding parts of the images.

Author reply:

We are sorry for the unclear images, and all the black/white images have been improved in contrast. (All the figures including Supplementary) The images correspond to the difference between the images with and without the pump pulse irradiation, and they are basically small in amplitude. We tried our best to enhance the contrast under the restriction that the original data is not changed. Please check the quality again.

The drawing of the photoexcitation regions was replaced to make it side-by-side with the image sequence. The shift of the drawing has made clear the comparison between the time-resolved images with the photo-excited regions.

We are sorry for the overlap with the time captions with Region 4. We moved the time caption position, and Region 4 has been displayed without an overlap. (All the image data figures, Fig. 1)

5) I recommend avoiding the use of red/green contrast in the graphs, which might challenge parts of the potential readers.

Author reply:

The color combinations in Fig. 3 and 4 have been changed in color. We suppose that the color visibility has been improved. (Fig. 2)

6) Figure 2/3/4: Here I suggest a different representation which also offers better comparability between the data collected for the complete Z scheme system and the single components. Further it will allow to avoid showing the action scheme twice. I suggest to prepare one figure containing all the data for direct comparison next to each other (i.e. Figure 2 panel a and Figure 3) and have another figure just containing Figure 4 and deleting Figure 2 panel b, which is not necessary.

Author reply:

Thank you very much for the advice for the improvement of the logical composition. We combined the scheme together to make it Figure 3. The temporal responses for the complete Z scheme and each component have been combined into a single figure with numbering.

(Fig. 2 and 3)

- 7) Could the authors shift the information on the hole quenching by methanol as proof for the nature of the second response in STOR arising from hole states mentioned on page 8 to the description of the data and assignment of the features on page 7? In my opinion this supports the flow of the logic in the presentation and discussion of the data. The data collected for STOR in presence of MeOH could even be added to Figure 3 panel b. This could nicely illustrate the point that the slower response 2 in STOR which is not present in the complete system is related to unquenched holes.

Author reply:

Thank you for the advice on the flow of logic. Surely there were some redundancies in the manuscript and revised the order of explanations according to your advice. The average responses for the Z scheme, BVOM only, and STOR only have been united in Fig. 2, and the scavenger effect by MeOH for STOR has been included in Fig. 2(c) (originally in Fig. S1 in SI). The scavenger effect was used for the assignment of the slow-rising component in STOR as the hole states, corresponding to Rh states.

(Fig.2(c), and p. 6 line 13-19)

- 8) The authors investigate the local responses and draw conclusions on the local activity of their sample. Are there microscopy images of the same areas of the sample so the spatial distribution derived from cluster analysis could be directly correlated with structural features visible under the microscope? Are there any inhomogeneities visible which could be directly related to the distribution of responses mapped out from the PI-PM investigation?

Author reply:

Thank you for the good advice, and we should have mentioned more details on the sample inhomogeneity, which would help readers to understand what caused the spotty responses. In most cases, the aggregates with several microns in diameter correspond to STOR. This is related to the original property of STOR particles, which was prepared by the solid reaction method and formed some amount of aggregates before mixing with BVOM, while BVOM particles are well dispersed. It is difficult to completely disperse STOR particles even after making the Z scheme samples.

The sample homogeneity is described in a sentence with additional data of SEM images, and EDS results for the photocatalytic sheet were added. You could recognize that BVOM was dispersed, but STOR formed aggregates.

(p. 10 line 21 – p. 11 line 4, p 15 line 2-4, Supplementary Fig. 3)

I think in Figure 5c, something in this direction is shown, please specify which microscopy technique this image is resulting from. In the text is mentioned the “original microscopic phase image”, I assume this means before excitation? Are there further images collected by other techniques which could give additional information on heterogeneity and distributions of defects in the printed system, etc. which could be related to the signals collected?

Author reply:

The reviewer is correct, and the “original microscopic phase image” corresponds to the optical image obtained by the same optical setup as the PI-PM method. As you mentioned, our optical resolution is limited, and we added more information by an SEM image and the corresponding EDS result.

The explanation of the optical image has been described and the sample information on the heterogeneity and aggregates is described in the text.

(p. 10 line 18-19, Supplementary Fig. 3)

The authors relate the structure which can be recognized in Figure 5c to aggregations. Do you think these are aggregation of all three materials, leading to active regions, and in the other regions the material is not properly assembled and hence shows no activity? Is there a way to show this by imaging the sample composition with some chemical contrast? Also considering the images of the other investigated regions I think there the correlation between aggregations

observed in the original phase image and the signatures from the analysis of the time resolved data is hard to see. From my point of view a convincing assignment, whether these aggregates are somehow related to a certain feature in the dynamics based on the presented data is not possible.

Author reply:

As the reviewer indicated, the description was not sufficient enough about the correlation between the aggregates and no activity regions. As we answered above, we added extra data for the photocatalytic sheet and clarified the origin of the aggregates. In these aggregates, we mostly observed the hole trapping to the Rh states (slow response $\sim 10 \mu\text{s}$), corresponding to the inactive sites for the water splitting. This was due to the chemical property of the STOR particles with strong aggregation, which could be improved by the synthetic and mixing conditions.

However, as the reviewer pointed out, the positions of the aggregates did not always match the positions of the slow-rising component. Since the optical image could provide the aggregates on the surface, the aggregates could possibly be buried in the film, and also, the mixing between STOR, BVOM, and ITO would not be sufficient.

Furthermore, the correlation between the slow-rising positions and the aggregates in absence of the ITO binder was not recognized, and it clearly indicates that the Rh states were generated by preventing the charge transfer between BVOM and STOR.

The sample specifications and the relation between the aggregates and the activity for water splitting were described in the text.

(Supplementary Fig. 3, p. 11 line 23-26, p 15 line 2-4)

9) SEM images are provided of the single components but not of the printed system. This would be interesting to get an impression on the heterogeneity of the water splitting sheet.

Author reply:

The SEM image for the photocatalytic sheet is provided in Supplementary, and the explanation of the sizes of the materials is described. Also, the evidence that the aggregates correspond to STOR has been provided by the EDS results.

(p. 10 line 21 – p. 11 line 4, p 15 line 2-4, Supplementary Fig. 3)

10) The authors relate the feature in the dynamics which is only present in ACN to the active regions of the sample. I agree, there are certain arguments for that. I think it would be interesting to use spatially resolved electrochemical microscopic techniques, e.g. scanning electrochemical probe microscopy (SEPM) to map H₂ evolution, on such a material to be able to actually correlate active and non-active regions with the results from the PI-PM investigation. This would give a real correlation between the observed spatial distribution of features in the dynamics and activity of the sample.

Author reply:

It is good advice, and we totally agree that the results need to be confirmed by photoelectrochemical measurements. Especially we have also paid attention to SEPM measurement, which potentially has a mapping of the photoelectrochemical activity. Actually, we plan to work on it, asking one of the researchers working on this measurement, but probably it will take more time than the revision period of this paper. We also consulted with a commercial measurement of SEPM, but it has a spatial resolution of 10 μm at the minimum and also costs a considerable fee for all the measurements.

Unfortunately, we could not make the measurements for comparison in this paper due to the above reasons. But we would surely work on it, and we would appreciate it if you could understand the situation.

Grammar issues to correct (list not complete)

11) Abstract last sentence: “This combination methodology could not only lead to the maximum performance of photocatalyst sheets but also applicable to other 28 systems involving electron transfer.” Something is missing here, please check this sentence.

Author reply:

The sentence was modified and simplified.

(p. 1 line 28-29)

12) Page 3 line 90/91: “...local mapping of the transient responses of photo-excited charge carriers. (pattern- illumination phase microscopy (PI-PM)). “

Author reply:

The sentence was modified.

Throughout the paper, the grammar and logical errors have been checked by commercial service.

Thank you again for the detailed advice to improve our paper.

Reviewer 2 comments

Reviewer: 2

Comments:

The authors studied charge carrier dynamics of Mo-doped BiVO₄ and Rh-doped SrTiO₃ samples in acetonitrile and water via a novel PI-PM technique. The method can spatially resolve carrier dynamics. By studying the materials in isolation and in a heterostructure in two different solvents, the authors were able to assign features in the PI-PM dynamic trace to carrier dynamic pathways. The work focuses on state-of-the-art materials and uses a novel, exciting, and interesting technique to learn about the underlying fundamental photocatalytic processes related to solar water splitting. Hence, the nature and scope of the work is appropriate for Nature Communications. I think many researchers in the field would find this study interesting. The data generally supports the authors' conclusions, but the manuscript requires two major revisions before publication. First, the authors qualitatively discuss the time scales and processes. Second, the authors should expand a bit on their discussion of how PI-PM works generally so readers who are not familiar with the previous 2 papers can understand the significance of the results. Hence, I recommend the paper be published after a major revision.

Author reply:

We would like to appreciate your evaluation of our paper and very happy to hear it. According to your advice, we made a phenomenological analysis of the processes to obtain the kinetics of this process. Due to several reasons, the analyses would not be perfect at this moment and need to be explored and now in progress. Also, additional information on the PI-PM method is provided for the readers who do not know this method, which would be most likely.

Major comments:

1. The impact of the work would be improved if the authors could develop a model to fit the time-dependent PI-PM data. This model would extract rate constants and provide deeper physical insight to the systems. Several models have been developed to quantitatively analyze time- or frequency-resolved data in the field of photoelectrochemistry (Peter, L. M.; Vanmaekelbergh, D. Time and Frequency Resolved Studies of Photoelectrochemical

Kinetics. In *Advances in Electrochemical Science and Engineering*; John Wiley & Sons, Ltd, 1999; pp 77–163. <https://doi.org/10.1002/9783527616800.ch2>). Since this is the third paper on the PI-PM technique, it seems appropriate that the authors would quantitatively extract information from the data.

Author reply:

Thank you for introducing a very instructive book chapter, which we have not studied. We read it through thoroughly and made an analysis for the time-resolved response. (Supplementary p.13-15, “*Phenomenological kinetic analyses for charge carrier dynamics in Z scheme materials*”) As you would see, the analyses have not perfectly completed yet, because there are many problems with the complete analyses. We have decided to use the phenomenological rate equation for the analyses, at least to obtain the range of the kinetic parameters and to know if the processes can be reproduced to explain the observation. However, we stopped the complete analyses to fit the model to our response due to the following reasons:

- 1. As the reviewer probably would understand, the diffusion processes were neglected because many physical parameters lack for complicated materials (doped, nanoparticles, and their combinations).**
- 2. The trap states and their distribution were neglected due to the same reason above, and energy level information cannot be considered.**
- 3. The depletion layer was not considered because it is not clear for particulate films and no-bias conditions.**
- 4. In most cases for particulate semiconductors, the lifetime of charge carriers cannot be described by exponential functions (due to the lifetime distribution of charge carriers for a variety of trap states), but we ignored the lifetime distribution for simplicity.**
- 5. The direct fitting for the data has not been tried; 1. because the response of the refractive index change consisted of several different responses, and as a result, there are many combinations of the parameters, and we could not decide which ones were correct, 2. because the S/N ratio of the transient responses was not good enough to decide the optimal fitting condition, 3. because the actual model needs to include the diffusion and lifetime distribution, but they were neglected in this analysis, 4. because**

the kinetic parameters depend on the positions, which is the key finding of this paper, and it is not appropriate to decide the parameter at this stage.

For the above reasons, we will study the full analyses in the next step of this study by using the responses with a high S/N ratio and full description of maths, and it would take more time to develop them.

At this stage, we showed here a simple phenomenological description of the processes to reproduce the typical responses for different components of the charge dynamics processes. We estimated the typical parameters to reproduce the responses and also showed the effect of a kinetic parameter of the charge transfer mediator, which is most important in this Z scheme system. The analyses have been described in the main text and Supplementary. Hopefully, this would help to understate for readers, and you could approve it.

(p.9 line 23-30, Supplementary, p.13-15)

2. The imaging capability of the PI-PM method is not clear because the PI-PM image is not compared to the region of interest in a side-by-side fashion. The reader has no idea if the signal is due to the noise or the presence of larger/smaller particle. The authors should correlate the PI-PM images with SEM images of the same region of the sample.

Author reply:

We are sorry for the unclearness of the correspondence of images and the region of interest. We made several improvements according to your advice together with the first reviewer. 1: Figure 1 has been updated to make it easier to compare the PI-PM images with the regions of interest. 2: the image contrast and brightness have been improved to make the images clearer. (However, the PI-PM images were 1-0.1 % change of the image pixel intensities, and obviously, there were limitations due to the SN ratio.) 3: the information of the aggregates (small particles) on surfaces has been given with an SEM image and the corresponding EDS data to show they were mostly STOR aggregates due to the chemical property of their strong adhesion. 4: The optical resolution of this PI-PM method was clearly mentioned; around 1 μm , and the smaller particles were difficult to resolve at this stage, which will be further improved under progress.

However, we could not spot the same measured regions by the PI-PM method by the corresponding SEM image. It was impossible to compare the measured regions by the two different equipment, which we would like to if we could find a good solution. We hope you could understand its difficulty.

(Fig.1 and all images)

(p. 10 line 21 – p. 11 line 4, p 15 line 2-4, Supplementary Fig. 3)

3. Figure 5-7 are essentially the same measurements on different samples and/or under different conditions. If the authors could quantitatively analyze the raw data and extract information, it would allow them to construct new knowledge, which could lead to new figures instead of repetitive figures.

Author reply:

They are the key figures in this paper. Although they look similar, they represent a clear indication of the difference of charge carrier behavior under the Z scheme condition, the Z-scheme without charge mediator, and water-splitting condition for the Z scheme condition. We believe that the original data and the clustering of charge carrier behavior, which are the most important data in our paper, must be in the main article for readers. However, as you mentioned, the image quality was not great, and they have been revised.

At the same time, as you suggested, kinetic simulations were performed to reproduce the temporal responses observed under different conditions, and they were described in Supplementary. We agree that they were important information, but the main focus of this paper should be given to the charge carrier mapping and not on the kinetics, which should be more focused when only the temporal responses are obtained and discussed. In our case, the spatial dependence of the charge responses is more important. Also, the SN ratio of the time responses was still not sufficiently high to make a proper kinetic analysis for these multiple complicated processes.

(All images, Supplementary, p.13-15)

4. When PI-PM is introduced in line 113, page 4, it would be good if the authors could explain the physical reason why the refractive index changes and where it occurs (e.g., in the solid, in the electrolyte, or at the interface). How is the refractive index change quantitatively

linked to the total number of charge carriers in the semiconductor? More explanation of the PI-PM technique in the main text would be useful here.

Author reply:

Thank you for the good advice to improve the readability. The origin of the refractive index change, in general, has been explained. Basically, the refractive index change is proportional to the number of excited charge carriers around the optical transition, and it is also valid for the Drude carriers (free electrons and holes). But the signal intensities cannot be compared for different types of charge carriers because the refractive index is a function of various physical parameters such as effective mass, permittivity, energy level, and so on. Furthermore, the charge transfer processes at the interfaces were frequently observed via the refractive index change, which has not been clearly described by the theory, but the dipole change at the interface could induce the phase shift of light and induce the refractive index change. This is the reason why various charge transfer processes have been explored by the detection via the refractive index, and this information has been described in the introduction.

The description was added to the text.

(p. 4 line 14-16, p.3 line 11-14)

5. For all the PI-PM images, can the authors show the SEM image of the same area of the sample? The images in Figure 5-7 are not very helpful.

Author reply:

As we have already answered in the previous comment, we revised our figures as possible we could. And also, we added an SEM image and the corresponding EDX result to identify the species on the surface. Although we would like to correspond the position of the measurement to the SEM images, it is impossible for these two different setups at this moment. Hopefully, we could recover the image much better by updating our equipment with a super-resolution microscope, and we are working on this.

(All images, Supplementary Fig. 3)

6. What is the fundamental spatial resolution of PI-PM versus what is being achieved in this study?

Author reply:

The same question was given by the first reviewer, and the optical resolution of images was $\sim 1 \mu\text{m}$. From the observation, a spot with a diameter of 500 nm could be recognized, but we could safely say that the resolution is less than $1 \mu\text{m}$ because sometimes we could observe a ring pattern for a small object, which is due to the diffraction.

The explanation is given in the text.

(p. 4 line 16)

Minor comments:

1. Page 3, line 92: what does “the image quality was recovered by the image” mean, exactly?

Author reply:

This is about our previous study. Since the image response is typically very weak, and we need to recover the image using the informatics calculations. Mostly we used two methods, one is the principal component analysis (PCA) based recovery, and the other is the least absolute shrinkage and selection operator (LASSO) based recovery. Although they could recover the signal contrast by reducing noise components, it could potentially remove small features in the images, and we did not use them in this study.

The explanation was added in the text.

(p. 3 line 26-27)

2. Is “ITO binder” in line 281 the best term?

Author reply:

We conventionally used the word binder, literary because it binds HER and OER materials, but the term is not general in scientific papers, and it has been revised to “ITO mediator.”

(Each part)

3. Line 311... define “local”.

Author reply:

The unclear word was removed to make the meaning clear.

Thank you very much for your valuable advice.

Reviewer 3 comments

Reviewer: 3

Comments

In "Charge Carrier Mapping for Z-scheme Photocatalytic Water Splitting Sheet by Categorization of Microscopic Time-resolved Image Sequence", Ebihara and co-workers report on an interesting method to visualize charge transfer on particles during water plaiting. The paper is well written and the provided data sufficient. Some comments are provided below:

Author reply:

We would like to appreciate your evaluation of our paper and very happy to hear you had a positive opinion on our paper. We revised our paper according to your advice, and we hope you would satisfy our revisions.

- On page 7 I could read "it is supposed that the holes in BVOM and electrons in STOR for water splitting were observed as a mixture in the response of the Z-scheme sample because these lifetimes matched by chance." Would it be possible to differentiate the process in one or the or the particle using laser excitations of various wavelength where only one of the two material is generating photo carriers?

Author reply:

It is a very nice point to make, and we would like to. However, the absorption of BVOM and STOR was difficult to select; BVO has an interband transition around 2.4 eV, and STOR have two absorption overlap from the Rh states, centered at 2.3 and 1.7 eV. Although the absorption of STOR extends to the red wavelength, the absorption is not related to the water-splitting. (valence band to Rh³⁺ state), and we cannot select a good absorption wavelength. This is the reason why we used each material for studying the response in the presence and absence of scavengers.

- Information about the origin of the energy levels responsible for trapping in "Step 2" should be included if known.

Author reply:

We could obtain information from the literature about BVOM and STOR, and summarized the information with references. Generally, with regard to STO, similarly as titanium oxide, a shallowly trapped state about 0.3 eV below the conduction band is believed to work as hydrogen generation, but there were no clear reports on the electron trap states used for hydrogen generation. The hole trapping states of BVO were also not much clarified yet, and some reports mentioned 50-100 meV higher than the valence band, but some of the other trap states are not active for water-splitting.

The explanation was added to the text.

(p.9 line 6-8)

- Could the charge transfer mapping (e.g. Fig 5.b) be assigned to any specific morphology or micro-structural defect in the particle?

Author reply:

As we replied to the other reviewers, the aggregates observed on the surfaces correspond to STOR, and it was confirmed by an SEM image and the EDS result, which have been included in Supplementary Fig. 3. In STOR, the charge carrier response was delayed about 10 μ s, which indicates that they have inactive for water-splitting.

They were additionally described in the text.

(p.10 line 21-p.11 line 4, p. 11 line 23-26, Supplementary Fig.3)

REVIEWERS' COMMENTS

Reviewer #1 (Remarks to the Author):

I am satisfied with the revision and recommend the paper for publication.

Reviewer 1 comments

Reviewer: 1

Comments:

Reviewer #1 (Remarks to the Author):

I am satisfied with the revision and recommend the paper for publication.

Author reply:

We would like to appreciate your evaluation of our paper and very happy to hear it.

Thank you very much for supportive comments.